# PLA Composites Reinforced with Flax and Jute Fibers—A Review of Recent Trends, Processing Parameters and Mechanical Properties

**DOI:** 10.3390/polym12102373

**Published:** 2020-10-15

**Authors:** Usha Kiran Sanivada, Gonzalo Mármol, F. P. Brito, Raul Fangueiro

**Affiliations:** 1Department of Mechanical Engineering, University of Minho, Azurém Campus, 4800-058 Guimarães, Portugal; francisco@dem.uminho.pt (F.P.B.); rfangueiro@dem.uminho.pt (R.F.); 2Mechanical Engineering and Resource Sustainability Centre (MEtRICS), University of Minho, Azurém Campus, 4800-058 Guimarães, Portugal; 3Centre for Textile Science and Technology (2C2T), University of Minho, Azurém Campus, 4800-058 Guimarães, Portugal; gonmrde@gmail.com

**Keywords:** green composite, biopolymer, natural fiber, compression molding, polylactic acid, flax, jute, tensile strength, impact strength

## Abstract

Multiple environmental concerns such as garbage generation, accumulation in disposal systems and recyclability are powerful drivers for the use of many biodegradable materials. Due to the new uses and requests of plastic users, the consumption of biopolymers is increasing day by day. Polylactic Acid (PLA) being one of the most promising biopolymers and researched extensively, it is emerging as a substitute for petroleum-based polymers. Similarly, owing to both environmental and economic benefits, as well as to their technical features, natural fibers are arising as likely replacements to synthetic fibers to reinforce composites for numerous products. This work reviews the current state of the art of PLA compounds reinforced with two of the high strength natural fibers for this application: flax and jute. Flax fibers are the most valuable bast-type fibers and jute is a widely available plant at an economic price across the entire Asian continent. The physical and chemical treatments of the fibers and the production processing of the green composites are exposed before reporting the main achievements of these materials for structural applications. Detailed information is summarized to understand the advances throughout the last decade and to settle the basis of the next generation of flax/jute reinforced PLA composites (200 Maximum).

## 1. Introduction

Composite materials have received much attention due to their versatile properties, which allow many applications in a huge number of fields. Composite materials are made up of two or more materials with considerably different physical and chemical properties. When combined, they make a material with properties that are different from the individual components. They are composed of a rigid phase, so-called matrix and the reinforcement, which is a strong load-carrying material. The strength and stiffness to support the structural load are provided by the reinforcement. The major benefits of the composite materials, when compared with conventional bulk materials, are their stiffness and strength relative to their typically low weight. These materials find applications in various industries like aerospace, automotive, sports, furniture, medical and packaging industries [1]. Various types of composite materials are developed with polymer matrix composites due to the flexibility, cost and ease of fabrication. Synthetic polymers are used in the production of these composites and are obtained from petroleum resources. The main limitation of these polymers is their non-biodegradability, which increases the amount of material that ends up in the landfills causing pollution [2]. The strict regulations from the government authorities also limit the use of traditional composite materials. These are usually made of glass, carbon or aramid fibers reinforced with epoxy, unsaturated polyester resins and polyurethanes. During recent decades, the scientific community has been looking towards sustainable materials to reduce its impact on the environment [3]. Therefore, there has been a constant interest in the development of green composites.

## 2. Green Composites—Historical Perspective

The composites materials prepared by reinforcing biopolymers with natural fibers are referred to as Green Composites. Coir, flax, sisal, hemp and so forth are examples of natural fibers, while starch and PLA are examples of biopolymers used for the preparation of Green Composites [4]. The development of the first green composites dates to the late 1980s. They are not only eco-friendly but also have a cost comparable to conventional composites. Green composites are not new to mankind and have existed in ancient times. For instance, they were used in the construction of the Great Wall of China. Rammed earth was used in the construction of walls during the Qin dynasty. The walls were built using red willow, reeds, gravel and sand during the time of the Han dynasty [5]. Mongolians in 1200 AD have used bows made with adhesively bonded laminates of animal horns and tendons, wood or silk. The usage of natural polymers is not new, as paper and silk were used in the past. From 300 B.C natural fibers were utilized as reinforcements where the straw was used as reinforcement in clay for the preparation of a composite bricks and they are in existence still today. [6,7]. It was in 1909 that Bakelite phenolic molding resin was invented and shortly after the wood flour or string and waste rages were introduced into brittle resin to form synthetic composites that were used in the radio and speaker cases [8]. In 1920s, Messrs Cladwell and Clay had used a natural fabric composite to produce airscrews for airplanes. Until 1930 not much interest was shown for the use of the synthetic composites in structural applications. By this time, a synthetic composite called Gordon Aerolite was made by De Bruyne. It was a composite manufactured from the hot pressing of unbleached flax yarn impregnated with phenolic resin. This material was used to produce the whole main wing spar for the Bristol Blenheim light bomber and the fuselage of the Supermarine Spitfire fighter aircraft [8].

After World War II the usage of natural fiber composites was significantly reduced as synthetic polymers were introduced from lower-cost petroleum-based polymers [3]. The search for lightweight and sustainable materials by aerospace manufacturers at the end of 1990 was concentrated on the natural fiber-reinforced composites [9]. Hence modern bio-composites appeared. However, the resin was synthetic and reinforced with natural fibers extracted from renewable resources. That is why they are also called partially green composites. In recent decades, to reduce the carbon footprint in the environment, much focus has been put on sustainable composite materials. Hence, the polymers extracted from the renewable resources known as biopolymers are used as a matrix material and reinforced with natural fibers to produce a fully “green composite material”.

## 3. Biopolymers

Several environmental issues such as waste generation, accumulation in disposal system and reproducibility were predominant with the use of many non-biodegradable polymers. As a result of using non-biodegradable polymers in composites, there was an increase in the volume of industrial and commercial dumps [10]. Hence much attention has been given to use biodegradable polymers as a matrix in the composite materials. These biodegradable polymers gained a lot of attention as they are designed in such a way that they degrade by the action of living organisms while conventional plastics do not. Therefore, research has been going on for decades to replace non-biodegradable polymers with biodegradable polymers. Biopolymers are produced from biomass resources such as corn, starch, cellulosic, proteins [3]. Classification of the polymers according to the origin and production is shown in Figure 1. Out of all the available biopolymers, the most commonly used biodegradable matrices are Polylactic Acid (PLA), Polyhydroxyalkanoate (PHA), starch and so forth [2]. These bio-based polymer matrices have good life cycle when compared to petroleum-based polymers [11].

Based on the new applications and demands, the usage of biopolymers is increasing day by day. Business Communications Company (BCC) research published a report in February 2019 and stated that global bioplastics market should reach 2.7 million metric tons by 2023 [12]. The capacity of biopolymer production was expected to increase from 2.33 million metric tons in 2013 to 3.45 million metric tons in 2020. Among the biopolymers, the focal products are PLA and starch-based plastics. Less amount of energy is required to produce PLA which in turn will release a low amount of greenhouse effect gases. The most encouraging biopolymer is PLA and it has been recognized since 1845 never was used commercially until the early 90s [13]. Now PLA is one of the most widely used polymers in green composites due to its mechanical properties and biocompatibility [11]. Moreover, PLA-based composites after utilization can be converted into water and carbon dioxide and can be utilized in the growth of agricultural products [14]. However, PLA degrades promptly in months at high temperature conditions (approximately 58 °C and 80–90% humidity) under aerobic compositing conditions [15]. In natural environment it may take several years for the complete disappearance of PLA [16].

### PLA

PLA is an aliphatic polyester with lactic acid as its basic constitutional unit. It was recognized since 1845 but was not available commercially till the early 1990s [17]. PLA can be synthesized by the polymerization of Lactic Acid (LA) or the Ring-Opening Polymerization (ROP) of Lactide [18,19]. Due to inherent disadvantages and environmental concerns, other new methods such as biosynthesis of PLA by enzymatic action and new solutions were developed that is, incorporation of non-toxic catalysts such as Magnesium (Mg), Calcium (Ca), Zinc (Zn) and so forth [20]. Figure 2 shows the PLA synthesis.

It is a renewable, recyclable, biodegradable and compostable polymer that shows outstanding processing ability. Due to its good properties Table 1, its applications comprise several industries, such as packaging, textile, biomedical, structural and automotive [13,17,21]. It is the only commercially available polymer with glass transition temperature (55–60 °C) above ambient and melting point (130–180 °C) [22] below the degradation temperature of lignocellulosic fibers (200 °C) [23].

Numerous distinctive characteristics, such as good transparency, high rigidity, glossy appearance of PLA have made it commercially and environmentally appealing. PLA has higher tensile strength and elastic modulus when compared to Polypropylene (PP), Polystyrene (PS) and Polyethylene (PE) [7]. There are also some limitations such as inherent brittleness and poor toughness and slow degradation rate which hinder its widespread application [13]. Biopolymers are compatible with many processing techniques such as injection molding, extrusion, compression molding, amongst others [19]. The various properties of PLA were as shown in Table 1 [24]. The mechanical properties of PLA are in comparison with PP, PS, High Density Polyethylene (HDPE), Polyamide (PA6) and are shown in Table 2.

Biodegradability is one of the significant aspects of sustainable materials. PLA is known to be entirely biodegradable. Through hydrolysis, microorganisms convert the LA into water and carbon monoxide. Biodegradation of PLA gets noticeable in two weeks after composting with other biomass such as compost soil and vanishes in 4 weeks [13].

PLA in its early degradation stage breaks down into LA monomers, where the ester bonds are cleaved hydrolytically. Microorganisms do the subsequent metabolic activity and the polymer breaks down into metabolic end products, such as water and CO_2_. Degradation mainly depends on the molecular weight and other factors such as temperature, time, impurities and residual catalyst concentration. The catalyst increases the rate of biodegradation of PLA and reduces the degradation temperature [13]. Figure 3 displays the life cycle of the PLA.

According to chemistry, polymer degradation can occur in three different ways (i) scission of the main chains (ii) scission of side chains (iii) scission of intersectional chains. The degradation of PLA happens mainly due to the scission of ester bonds. Factors such as oxidation, photodegradation, thermolysis, hydrolysis, biodegradation or enzymolysis. New kind of techniques such as biochemical process (chemical hydrolysis and biodegradation in natural soil microcosm), microbial degradation (actinomycetes, bacteria and fungus) and enzymes such as proteases, lipase and cutinase are reported to be studied for the degradation of PLA [16]. It was reported in one study made by Yanbin Luo and his team that the addition of nanoparticles (TiO_2_) has improved the biodegradation rate under controlled composting conditions [26].

Significant modification of PLA is required for improving the properties such as mechanical strength, heat distortion temperature, barrier properties and durability. The most commonly used methods of polymer modification comprise of chemical co-polymerization, polymer blending and nanocomposite technology and are well reported in the literature [15].

## 4. Natural Fibers

Natural fibers are renewable resources that may be obtained in a sustainable way. Eco-friendly nature, low cost, renewability, local availability compared with synthetic fibers have made them as potential alternative materials in the composite industry [27]. Utilization of plant-based natural fibers such as flax, jute, sisal, ramie, coconut coir, pineapple leaf fiber, kenaf, bamboo fiber and so forth have improved due to the increased efforts made by researchers [28]. Table 3 shows the classification of natural fibers. The chemical composition of natural fibers is shown in Table 4. Natural fibers with high cellulose content have the higher values of strength.

Most natural fibers are composed of cellulose, hemicellulose, lignin, waxes and several water water-soluble compounds. Table 4 enlists the chemical composition of different natural fibers. The strength and stiffness of a natural fiber depend on the amount of cellulose in it [29]. Mechanical properties also depend on the microfibril angles and the degree of polymerization of cellulose [13]. Moreover, ecotype, maturity and location of the plant and fiber extraction process influence the properties of natural fibers [13,30]. The different factors influencing their quality are shown in Table 5. The selection of the fiber mainly depends upon the strength requirements for the various applications. Table 5 lists out different physical and mechanical properties of the natural fibers reported in the literature. The important parameters that could establish the properties of the fibers are density, microfibril angle, Young’s modulus and fiber elongation [31]. Because of the combination of the environmental and economic benefits, the natural fibers are emerging as possible alternatives to synthetic fibers to reinforce composites in various applications [32].

The information compiled in Table 5 demonstrates that the mechanical properties of the natural fibers are lower when compared to synthetic fibers. However, the properties of the natural fibers can be enhanced and can be made comparable with the values of synthetic fibers by proper surface treatments. The value of Young´s modulus for some natural fibers is in the same order of magnitude of glass fibers. The lower density and specific modulus of the natural fibers has attracted the interest of the industries and becomes a potential candidate for the applications in green composites [3].

The advantages of natural fibers are their low density (1.25–1.50 g/cm^3^), good specific mechanical properties, recyclability, biodegradability, abundant availability and so forth [28]. The other significant factor is the lower price of natural fibers (200–1000 US$/ton) when compared to the glass fibers (1200–1800 US$/ton) [38]. All these features led industries to consider the use of natural fibers as reinforcements for the manufacturing of Polymeric Matrix Composites (PMC). However, there were some limitations to use natural fibers as reinforcements in PMC such as their poor wettability, incompatibility with some matrices and high moisture absorption. The major advantages and disadvantages are listed in Table 6.

The most common drawback of natural fiber composites is the lack of proper fiber-matrix adhesion. Poor adhesion between fiber and matrix can lead to lower mechanical properties. Hence it is essential to have a good bonding between fiber and matrix. The full capabilities of a fiber-reinforced composite cannot be exploited with a deficient adhesion at the interface. Nevertheless, it is possible to improve these properties with the help of physical and chemical treatments [39].

Another major drawback inherent to natural fibers is their hydrophilic nature [13]. To reduce the effects of the hydrophilic nature of the natural fibers a great range of chemicals, physical and biological treatments are available. Examples of the treatments include silane, acetone and alkali treatment, which have shown positive results as per the literature [14]. These treatments reduce the impurities on the fiber surface and help to decrease their hydrophilicity. This helps to enhance the fiber/matrix compatibility, hence improving the properties of the Natural Fiber-Reinforced Composites (NFRC) [13].

Natural fibers, to be used as fiber reinforcements in polymer matrix high cellulose content and low microfibril angle are essential [42]. From Table 4 and Table 5 it can be observed that natural fibers such as flax, hemp, jute, pineapple and ramie have high cellulose content and low microfibril angle. The respective fibers have good values of tensile strength. While cotton has high cellulose content (82.7%) its tensile strength is not as good as other fibers due to its high microfibril angle (20–30°). Flax and jute have the combination of high cellulose value and low microfibril angle that gathers the attention to study about these fibers as reinforcements in PLA matrix.

## 5. Treatment of Natural Fibers

Along with the advantages such as low density, low cost and biodegradable, natural fibers also have some drawbacks when used in composites. Poor compatibility with different matrices, high moisture absorption and swelling lead to the formation of cracks in brittle matrices [43]. The poor compatibility of the fibers with polymeric matrices is due to the presence of the free water and hydroxyl groups, especially in the amorphous region. Hence the fibers are subjected to various treatments [44]. Since natural fibers have a high cellulose content, the treatments seek to modify the structure of the cellulose by substituting the hydroxyl groups by some chemical groups. The substitution groups play the role of plasticizer and improve the thermoplasticity of the composites [44].

The physical methods such as stretching, calendaring, plasma treatment, the electric discharge modify the fibers, thus improving the mechanical performance of composites. The mechanical bonding of the polymers is influenced by the change in the structural and surface properties of fibers as a result of the physical treatments. The chemical composition of the fibers is not extensively changed by the physical treatments, so the interface improvement is due to the increased mechanical bonding between the fiber and matrix [45]. Plasma treatment is one of the most often used method. Plasma is partially ionized gas with highly excited atomic, molecular, ionic and radical species with free electrons and photons. This ionized gas is produced by applying an electric field over two electrodes with gas in between either at atmospheric pressure or vacuum. The gases such as oxygen, nitrogen, helium and air can be utilized in this process. In this method various functional groups are introduced and these functional groups can form strong covalent bonds with the matrix and leads to the improvement in the fiber-matrix interface. It can also improve the surface roughness of the natural fiber that results in the good mechanical locking attachment between the matrix and the fibers [43]. This treatment can also generate free radicals that can react with the oxygen or other gases and eventually leads to the generation of surfaces with difference character (hydrophobic/hydrophilic).

Chemical methods such as alkalization, acetylation, benzoylation, bleaching and silane are used to improve the fiber surface. The researchers were successful and improved the adhesion characteristics of the fibers, thus improving the mechanical properties of the composites. From the literature available, the chemical treatments have removed the hemicellulose, lignin, pectin and oil contents from the fibers. Because of the removal of impurities, an improvement in the surface roughness was achieved, hence improved the fiber-matrix adhesion. The improvement of the properties also depends upon the type of treatment used and when used chemical methods concentration and duration of the treatments also influenced the results [46]. Most of the researchers have used one type of chemical treatments but a combination of treatments was also used in their works and observed sensible results [47]. It is evident from the literature that these treatments have achieved various levels of success in the improvement of the fiber-matrix adhesion in natural fiber reinforced composites [44]. Alkali and silane treatments are most often used chemical treatments.

Alkali treatment is also known as mercerization and it involves the treatment of the fibers with sodium hydroxide that result in partial removal of lignin and hemicellulose and complete removal of pectin, wax, oils and other organic compounds [48]. The reaction of sodium hydroxide with the natural fiber (Cell–OH) is thought to occur as shown below as Equation (1):Cell–OH + NaOH → Cell– O-Na⁺ + H_2_O + Surface Impurities(1)

Reaction between the NaOH and functional groups of the natural fibers [44].

The removal of the cementing materials results in the better packing of cellulose chains and hence there is an increase in the percentage of crystallinity index. Also, this treatment results in the decrease of spiral angle and increase in molecular orientation, improving the elastic modulus of the fibers. More cellulose is exposed after the removal of the impurities from the surface that gives rise in the additional sites for the mechanical interlocking and it is thought to be responsible for better fiber-matrix adhesion. The factors such as concentration of the NaOH, time of the treatment and the temperature plays important role in achieving the optimum characteristics of the fibers [44].

Silanes are one of the promising agents for the improvement of the chemical interactions between the fiber and the polymeric matrix. The silane molecule has bifunctional groups, which react in two phases to form the bridge between them. The reaction is shown in Figure 4 below.

Silane coupling agent hydrolysis is the first step in the reaction mechanism. The alkoxy groups present in the silane molecule are hydrolyzed to generate reactive silanol groups. Later, the hydrolyzed silane solution is mixed with the natural fibers. In this stage, reactive silanol groups in the silane bind to the OH groups present on the fiber surface by hydrogen bonds. The alkyl groups on the other side of the silicon connect to the polymer functional groups forming a siloxane bridge between polymer and the fiber. This results in the increase of the hydrophobic character of the surface and improvement in the strength of the interface in the matrix [48].

## 6. Composite Processing

There are several methods for producing the composites such as compression molding, extrusion, injection molding, filament winding, resin transfer method, vacuum infusion method. In this report, focus was given to the production of composites by compression molding.

### 6.1. Compression Moulding

Compression molding is generally used for thermoplastic matrices with loose chopped fiber or mats of short or long fiber either orientated or aligned. Heat and pressure are applied after stacking fibers and thermoplastic matrix sheets. Good quality composites can be obtained by controlling the production parameters such as viscosity, pressure, holding time and temperature, depending on the type of matrix and fiber [49]. Composites production through film stacking reduces natural fiber degradation as it involves one temperature cycle [50]. Care with temperature is essential as the difference between the processing temperature of the matrix and the temperature at which the fiber degradation initiates is relatively small. Above 200 °C, a reduction in fiber strength may be perceived [51]. Hence a compromise must be made between wettability and avoiding fiber degradation. The results from a study, which explored a production range of temperatures, yielded 150 °C as the optimum temperature for obtaining good tensile values for Polyester/Flax amide composites [52]. Flexural properties were found to be less independent below 150 °C but reduced significantly beyond 200 °C. The optimum temperature of jute yarn and bacterial co-polyester Biopol was found to be 180 °C for obtaining optimum mechanical properties [53]. For non-woven mat reinforced with PP samples produced at around 200 °C recorded a good strength [54].

Sheet molding is an alternative process to the film stacking method [55]. Some of the processing parameters for making PLA/Flax using compression molding, combination of hot press and injection molding and combination of extrusion and injection molding are shown in Table 7, Table 8 and Table 9 respectively. Processing parameters for manufacturing of PLA/Jute composites using compression molding and combination of process extrusion and injection molding are shown in Table 10 and Table 11. Figure 5 shows the compression molding set up for producing composites by using film stacking method and using pre compounded fiber matrix. The pre compounded fiber matrix can be obtained via extrusion where the fiber and matrix are mixed and pellets or prepregs are formed.

### 6.2. Compression Moulding of Precompounded Mixture

Through this process, a pre-compounded fiber-matrix mixture is hot pressed to produce a desired product. Few researchers have reported using this method for the fabrication of bio composites. Several researchers reported the use of extrusion and melt binders for the preparation of fiber-matrix compounded mixture before the use of compression molding [81]. Other researchers used a two-roll mill was used to obtain the pre-compounded sheets and those sheets were hot pressed at 20 MPa and 170 °C for 4 min and subsequent cooling at room temperature at 5 MPa to produce PLA/Jute and PLA/Ramie composites [82]. In the case of PLA/Coir composites, specimens were produced by hydraulic pressing. PLA sheets were fabricated by placing PLA pellets in an oven at 180 °C for 15 min, followed by applying pressure of 1 MPa in between aluminum sheets with the help of hydraulic press for 15 min at room temperature and subsequent cooling for 30 min. Pre dried coir fibers were then stacked between PLA sheets and heated in oven for 15–30 min and pressed at 1.5 MPa for 15 min. The laminates were cooled at room temperature for 30 min [83].

### 6.3. Film Stacking

Film Stacking is a simple processing route for the fabrication of laminates. The matrix films produced from the biopolymer and fiber mats (woven, plain or randomly aligned) are stacked alternatively and compressed between the heated mold plates [81]. Film stacking technique is conventionally used to develop PLA/Jute fiber composites. According to the literature, PLA films of 0.2 mm thickness were produced with the help of extruder. 40 wt % jute fiber mats were stacked in between films in a frame/mold and was pre-compressed at 3.3 MPa for 15 s. Pre-compressed stack was then placed between the heated platens under pressure of 400 Pa for 3–10 min at temperature of 180–220 °C [84]. In order to produce PLA-hemp bio composites, PLA pellets were converted into films by using an extruder while a hand-carding machine was used to align the industrial hemp fibers. PLA films and fiber mats were stacked alternatively in a mold and pre-compressed for 5 min at a temperature of 185 °C at a pressure of 2 MPa. The pressure was increased to 5 MPa thereon and allowed to cool at room temperature [85,86]. In another study, PLA-Sisal based bio composites of 4 mm thickness were produced by using film stacking method. PLA pellets were compressed to form 1 mm thick films and then fiber mats were stacked alternatively within the mold. The stack was compressed at 4 MPa for 8 min at 180 °C. The pressure was then increased to 6 MPa and allowed to cool at room temperature [87,88].

PLA-kenaf bio composites, specimens were obtained where initially PLA pellets were converted into 1mm thick PLA sheets using a compression molder at 190 °C. Subsequently layers of fiber mats and PLA films were stacked and compressed at 4.8 MPa at 190 °C for 12 min and further compressed at 11.7 MPa for 5 min and cooled at room temperature [89]. Different authors explore the use of short jute fibers of 10–15 mm to reinforce with commercially available PLA films of 0.3 mm thickness to produce PLA/Jute composite by using film stacking hot pressed method. PLA films and jute fibers with varying volume fractions were stacked alternatively between compression molder at a temperature of 170 °C for 10 min at pressure of 1.3 MPa [90].

### 6.4. Fiber Orientaiton and Distribution

Regarding the study of the orientation and distribution of the fibers, the composite displays better mechanical properties when the fiber is aligned parallel to the direction of the applied load [91]. Depending on the matrix viscosity and mold design, some alignment is achieved during injection molding. It is a hard task to achieve fiber alignment with natural fibers compared to continuous synthetic fibers [92]. However, long natural fiber can be carded and positioned manually in the form of sheets prior to matrix impregnation. Wrap spinning was a method in existence since 1970 for producing aligned fiber yarns. It is a method where short fiber can be converted to continuous filaments by using a continuous stand wrapped around discontinuous fiber with enough frequency to provide the required integrity for subsequent processing. It was reported that the aligned fiber yarn improves tensile and flexural strength and stiffness over conventional twisted yarns [91]. Another approach that has been mentioned is the Dynamic Sheet Forming (DSF), a method to align fiber traditionally in paper production. In this method short fibers are suspended in water and this solution is sprayed on to the rotating drum through nozzle. The rotating drum is covered with a wire mesh, which helps in removing water and brings about the alignment in the spray and rotating direction [52].

In a study it was found that the mechanical parameters of a composite alfa-polyester decreased with the increase in the angle of fiber orientation [93]. In another study, it was found out that strength and Young’s modulus were influenced by the fiber orientation. The tensile strength (72.75 ± 6.26 MPa), (34.75 ± 4.63 MPa), (22.01 ± 3.38 MPa) and Young´s modulus (8.77 ± 1.44 GPa), (4.62 ± 1.04 GPa), (3.70 ± 0.65 GPa) were found to be decreasing with the increase of the angle of fiber orientation that is, 0°, 45°, 90° respectively [94].

The specimens prepared by compression molding using film stacking method have the best control to the designer over fiber orientation within the bio composites. According to the load-bearing requirement, continuous fibers can be woven at different angles. The use of carding process has been reported in literature review prior to the film stacking method. In bio composites developed using the film stacking method, fibers aligned in the direction of applied load have the more load bearing capacity when compared to the randomly aligned short fiber reinforced (pre-compounded) bio composites [81].

## 7. PLA/Flax Composites

Melt compounding and solution dipping process were employed to produce PLA/Flax composites. The effects of different surface modifications techniques such as alkaline, silane, polymer coating + alkaline and polymer coating + silane on the mechanical performance were studied. Polymer coating on the surface of the flax fibers combined with silane treatments showed an improvement in the performance. The maximum tensile strength of 74.3 MPa and impact strength of 12.7 kJ/m^2^ were observed in the salinized and polymer-coated flax fiber composites [67]. Polymer coating also showed a decrease in the hydrophilicity nature of the flax fibers. Similar composites were produced by using a twin-screw extruder, reporting higher values of tensile modulus (3.8 GPa), impact strength (15 kJ/m^2^), elongation at break (6.2%) for PLA/Flax composites when compared to the neat PLA. The toughness of the PLA/treated fiber composites has been increased when compared to the PLA/untreated fiber composites [63].

Comparing the properties of two types of unidirectional flax composites using PLA as matrix: one made with layers of aligned flax rovings alone and the other containing an additional paper layer fabricates using paper making techniques, it was observed that the PLA/Flax composites have better performance in terms of impact strength (800 ± 15 J/m), tensile strength (339.0 ± 22 MPa) and flexural strength (363 ± 26 MPa) than PLA/Flax-Paper composites. The specific tensile properties of both the composites can be compared with the composites made of woven glass fabrics impregnated with epoxy. It also observed that PLA/Flax-Paper composites have achieved good impact strength (600 J/m) [64].

When flax was grafted, composites produced by hot press molding and gel dip coating technique exhibited a significant improved adhesion bonding of the modified fibers to the matrix. This led to an increase in the impact resistance of the PLA when reinforced with fiber by three times and observed to be 15.4 kJ/m^2^. The hydrophilic nature of the fiber was reduced by 18% in the modified grafted fiber composite [65]. In order to evaluate the effect of fiber treatment, three different approaches were used to produce PLA/Flax composites by film stacking. The fibers were treated with Maleic Anhydride (MA), Silane (ST) and Tributyl Citrate (TBC). In any case, the addition of fibers increased the mechanical properties. The tensile strength, Young´s modulus and flexural strength were improved by 80%, 300% and 27% respectively when compared to the neat matrix. The highest values of tensile strength (102.5 ± 5.2 MPa), Young’s modulus (25 ± 1.4 GPa) and flexural strength (140 ± 6.9 MPa) was obtained for the composites treated with silane, maleic anhydride and untreated respectively. The other surface treatments also improved tensile and flexural properties [62]. Later, the flexural properties and morphology of the surface of the composites were examined and found that 2% *w*/*w* silane content improved the flexural strength of the composite material by 18% [57].

Nano-Coated Flax was reinforced with PLA to produce composites by hot pressing. The values of ultimate tensile strength, tensile modulus and interlaminar strength were found to be (187 ± 10.6 MPa), (12.2 ± 0.33 GPa) and (24.84 ± 0.650 MPa) respectively. Ultimate tensile strength, tensile modulus and interlaminar strength was increased by 6%, 13% and 20% respectively when compared to the raw composites. Coating the fibers also allowed the decrease in water absorption by more than 10% in comparison with raw composites and their protection during conditioning, preserving their mechanical properties [61]. With the aim of improving flame retardancy, a thin coating of polydopamine adhesive was used to protect hot compression molded composites. Limiting Oxygen Index (LOI) values were highest for the Fiber-Modified with Iron Phosphonate (Fep-Flax) composites and increased by 26.1% compared to pure PLA. The self-extinguished ability of PLA/Fep-Flax composites improved remarkably and decreased the combustion time. PLA/Fep-Flax showed reduction of tensile strength (55.4 ± 0.9 MPa) by 6% when compared to neat PLA, however there was an improvement of Young´s modulus (2908 ± 104 MPa) by 35% [58]. Table 12 the values of the mechanical properties obtained by various researchers.

The influences of the fiber alignment were assessed by comparing the performance of Random (NM) and Unidirectional (UD) fiber composites produced by compression molding. Air laying was used to produce random flax reinforcements. UD flax composites showed improved properties when compared to NM flax composites. The tensile(151 ± 7 MPa) and flexural strength (215 ± 17 MPa) of PLA/UD Flax composite was increased 82 and 65% respectively and the tensile (19 ± 2 GPa)and flexural modulus (19 ± 1 GPa)was improved 99 and 90% respectively compared to NM PLA/Flax composite. The PLA/UD Flax and NM PLA/Flax lost 19 and 27% mass respectively after 120 days as per soil burial test [60].

Although PLA/Flax composites already exhibit excellent properties, it is possible to improve this material with the inclusion of other types of fibers. This process that combines more than one type of fiber is known as hybridization. In this sense, a first attempt was made to compare the properties of PLA/Flax composites and PLA/rayon produced by injection molding. Some of the mechanical properties of PLA/rayon samples were better than PLA/Flax composites at a fiber mass proportion by mass of 30%. The highest impact strength and tensile strength of 72 kJ/m^2^ and 58 MPa, respectively, were observed for rayon reinforced with PLA. However, the highest Young’s modulus (6.31 GPa) was found for the composite made up of flax reinforced with PLA. Poor matrix fiber interfacial bonding is found in both cases [66]. From this study was concluded that a potential benefit could be obtained by the hybridization of PLA using both types of fibers. The hybridization prospective of PLA-based composites were assessed by producing compression-molded samples reinforced with Cotton Gin Waste (CGW) and flax fibers where overall fiber fraction is 30%. The flexural modulus increased by 42% by the addition of 30 wt % of CGW compared to neat PLA and there was a reduction of flexural strength by 85% in comparison with neat PLA. Nevertheless, in the hybrid composites the addition of 10–20 wt % of flax fiber reduced the flexural properties. The modulus of elasticity for the composites containing 30 wt % of CGW and composites containing 30 wt % of flax fibers were comparable [56]. In this regard, hybridization produced insignificant advances given the similar nature of the two types of reinforcement.

## 8. PLA/Jute Composites

After the initial studies, long jute fibers (6 mm) were incorporated up to 50% by mass into PLA, which resulted in the enhancement of flexural strength and modulus of injection molded composites, although impact strength of the composite did not increase [76]. The effect of long (618 mm) and short (387 mm) jute fibers on the performance of PLA injection-molded composites was studied, the samples made of short fiber fillers showed optimal performance, exhibiting a significant 182% increase in strength (90.7 ± 1.3 MPa) compared to that of pure PLA%. The effective dispersion of the fibers and the high intensity mixing of the fibers have led to the improvement in the interfacial strength which leads to the improvement of the properties of short fiber fillers. The suppression of hydrolysis was necessary for the improvement of overall performance for both long and short fibers [77].

Another possibility of producing PLA/Jute composites is the combination of both PLA and jute filaments to produce composite spun yarns to be aligned in the same direction. this type of composites were produced by pressing the spun yarn at different temperatures [70]. From the results the optimum compression molding temperature was found to be 185–195 °C for fabrication the jute spun yarn/PLA unidirectional composite. It was clear that the impregnation quality and dispersion of fiber bundle increased with higher molding temperatures, since the elastic modulus increased. While increasing the molding temperature, the achievement ratio of tensile strength was decreased because of deterioration of jute fiber [70]. Jute also leads to an improvement of the PLA mechanical performance when its strands are added. In this regard, when strands of jute act as reinforcing elements, lignin content plays a key role. Hence, PLA composites were produced by injection molding where treated jute strands are used as reinforcements and the amount of lignin present in the jute are varied. The obtained results showed that mechanical properties of PLA can be interestingly improved with the incorporation of 30 wt % of jute but in different magnitude depending on the chemical composition thereof. A lignin mass fraction of 4% was found to be most suitable as PLA reinforcement because of the improved mechanical properties. There was an enhancement of tensile strength up to 46% as the amount of lignin was decreased, as the interface between fiber and matrix was improved [95].

When woven jute fabric is introduced to reinforce PLA, it is possible to increase the impact strength of the plain polymer as well as the tensile strength with only a minor reduction of the flexural modulus [96]. To assess the low-velocity impact behavior of PLA/Jute laminates with alternated fabric layers 0°/90°, symmetrically arranged were developed with respect to the middle plane of the laminate in between PLA and pressed them. The obtained composites were tested by a falling dart machine operating at impact energies equal to 5 J, 10 J and 20 J. The results revealed that the composites were able to resist the first two energy levels with a barely visible damage. Minor fiber-breakage and delamination phenomena were registered by photographic images taken on the back side of impacted samples. Only impact tests at 20 J, gave rise to severe impact damage of bio composite samples with penetration of the impactor and delamination phenomena apparently confined to the material-impactor contact area [96]. Compared to other fibers with high cellulose content like cotton fibers, the tensile modulus of the PLA/Jute composites is approximately 25% higher than that of the PLA/cotton composites [75]. Also, the compressive modulus of the PLA/Jute composite was 38% higher than that of the cotton composite. However, the compressive and tensile strength of the cotton/PLA composite was better than that of the PLA/Jute composite [75].

As previously described, despite all the gains of using jute to reinforce PLA composites, these fibers may undergo hygrothermal aging from the combined effects of humidity and heat. In a study it was observed that in composites produced by injection molding three degradation stages of hygrothermal aging of PLA/Jute composites take place and it can be related to the changes in mechanical properties [80]. In stage I, plasticization due to water absorption increases the ductility with a reduction in strength and tensile elastic modulus. In stage II, a detachment between the fiber and the matrix due to differential swelling leads to the weakening of the interface with a significant, detrimental effect on the mechanical properties. In stage III, the hydrolysis of the matrix causes microcracking with a further significant decrease in ductility. The durability of short jute fiber reinforced PLA composites in distilled water at different temperatures was studied. From the results of the water absorption test, it was noted that the composites maintained at 23 °C, 37.8 °C and 60 °C followed the Fickian behavior in the first stage. Once the rate of weight gain (Mt) was reached, the saturation point after Mt of PLA/Jute composites is like that of pure PLA [97]. However, the composites at 60 °C deviated from the Fickian law. PLA has shown slower water absorption rate compared to the PLA/Jute composites [97]. The mechanical properties have shown a significant decrease at the beginning of aging, then stabilized for a time period and decreased at last stage. These 2 researches [80,97] regarding aging mechanisms of PLA/Jute composites reported several undesired effects such as plasticization of the matrix, swelling of materials, structural damage of the PLA/Jute composites, change of the PLA crystallinity and hydrolysis of jute fiber and PLA matrix. Therefore, a fiber treatment is convenient to sort these drawbacks out. Table 13 shows the values of mechanical properties determined by various researchers through their experiments.

The most reported treatment in the literature about natural fibers in polymeric composites is alkali treatment. The effects of the treatment on both the single jute fiber and the PLA/Jute composites are examined and results revealed that the structure and shear strength of jute fibers and their composites were strongly affected by the surface treatment [98].

Moreover, the interfacial shear strength of the composite generally increased with an increase in the treating time and the concentration of NaOH solution and the maximum value was obtained at 8 h treatment with 12% NaOH solution. Meanwhile, alkali treatment of jute fibers significantly improved the tensile properties of both single fibers and composites. The optimum tensile properties of PLA/Jute composites were obtained at a 15 wt % fiber content and a processing temperature of 210 °C. Both the maximum flexural modulus and strength of composites were obtained at 220 °C and 15 wt % fiber content [98]. The influence of the alkali treatment with NaOH at various concentrations (5%, 10% and 15%) and with H_2_O_2_ was evaluated [99]. Fibers were added in different weight concentrations (5%, 10%, 15%, 20% and 25%) of untreated and treated jute fibers in PLA matrix. In composites produced by injection molding, it was observed from the experimental results that the surface modification of jute fibers with NaOH improved the flexural properties (tensile strength, Young’s Modulus and density) [99] but decreased the impact resistance [100]. It was found that the water absorption got leveled off after 24 h and was increased with the fiber content. Alkali treatment have reduced the water absorption rate and make fibers as hydrophobic.

In addition to alkali treatment, it is possible to apply a subsequent silane treatment to further improve the properties of PLA/Jute composites. The effects of 2 different treatments on the fiber properties were compared using (i) alkali (NaOH) and (ii) alkali followed by silane (NaOH + silane) separately [106]. It was found that the surface treatments led to the improvement of thermal stability and crystallinity index of jute fibers. Jute fibers treated with (NaOH + silane) showed lower crystallinity index compared to NaOH treated fibers due to the conversion of crystalline cellulose into the amorphous cellulose as a result of silane of the jute fibers. Matrix-fiber adhesion between PLA and jute fibers, which was examined in terms of Interfacial Shear Strength (IFSS), was found to be highest for (NaOH + silane) treated jute fibers [106]. When these treatments were applied in jute reinforced PLA composites with 30 wt % of reinforcement [101], It was reported that the tensile strength of the reinforced composites was increased by 5% compared to neat PLA. PLA reinforced with jute fibers treated with NaOH aqueous solution showed an improvement of 10% in comparison to neat PLA. Furthermore, an improvement of 32% was seen in the composites reinforced with NaOH + Silane treated jute fibers (PLA/JFNASI) [101]. Impact strength of PLA/JFU, PLA/JFNA and PLA/JFNASI was shown improvement of 34%, 49% and 56% when compared to neat PLA respectively [101]. The addition of jute fiber has increased the storage modulus and it was further improved with the surface treatments. It is also possible to apply physical treatments, as electron beam, to improve the thermal properties of the PLA/Jute composites. It was reported that PLA/Jute compression-molded composites presented an increased thermal stability, storage modulus and heat deflection temperature with an electron beam irradiation of jute at 10kGy [71].

In order to improve the flame retardancy of PLA/Jute composites, phosphorous-based compounds exhibit excellent fire resistance for polymers and their composites. A compound was synthesized by the reaction between 9,10-Dihydro-9-oxa-10-phosphaphenanthrene-10-oxide (DOPO) and maleic acid (MA) and was used in the composite formulation, ensuring a good interface by providing active sites linked between DOPO and PLA or jute [79]. Moreover, the addition of DOPO-MA improved the tensile, flexural and impact strength when compared to that of incorporation of DOPO. Flame retardancy was also improved and achieved due to the formation of char layer to protect the inner composite from further burning [79].

Given its chemical composition (Table 4), jute can be used to obtain Crystalline Cellulose (CC) to reinforce PLA composites and it was reported that the thermal stability and the crystallinity of the composites have increased when composite was prepared from CC and PLA [74]. It was observed the existence of hydrogen bonding between CC and PLA when the composite was fabricated by extrusion or hot press. Moreover, PLA/CC composites showed increased hardness and tensile strength as well as better antimicrobial properties [74]. In a similar experiment, Nano Fibers (NF) from waste jute fibers were used to reinforce PLA films by solvent casting. For samples with 5 wt % of NF, the initial modulus (3.30 ± 0.05 GPa) and tensile strength (70.30 ± 6.32 MPa) were increased by 217% and 171% respectively compared to neat PLA films. The composites with a 10 wt % of NF showed a decline in the mechanical properties, however they have showed a 16% and 62% improvement in crystallization temperature and crystallinity respectively. Mechanical properties were improved until 5 wt % fiber content and declined thereon. Huge improvements of storage modulus by 475% was observed at 60 °C [103]. The addition of different nanoparticles, such as nano-silica, enable the formation of rigid-soft core–shell structures to improve the mechanical properties of woven jute fiber-reinforced PLA composites. Press-molded composites with rigid core and soft-shell nanoparticles reinforced with jute fibers were produced and results indicated an increase in tensile strength (from 43.1 to 52.4 MPa) compared to composites without nanoparticles [105]. The addition of SiO_2_–NH_2_ and SiO_2_–PBA–NH_2_ modifiers further improved tensile values to 71.9 MPa and 77.5 MPa respectively, while flexural strength and modulus were increased by 38% and 101% when compared to Natural Fiber-Reinforced Polylactide Composite(NFPC)respectively [105].

## 9. Trends in Green Composites

The next generation composites are developed with inclusion of special functionalities such as high strength, stiffness or toughness, fully transparent, autonomously self-healing, fire retardant, nanocomposites for biomedical applications, resins-less composites, light-weight foam composites, composites with gas barrier properties [107]. In the attempt to produce the highest strength and stiffness among composites, a critical task is to produce of fibers which have these characteristics. Research at Groningen University was successful in producing high strength cellulose fibers. It was achieved by dissolving pure cellulose in highly concentrated phosphoric acid to form Liquid Crystalline Cellulose (LCC) and then using the air-gap wet spinning method to spin the fibers. the results showed that it attained strength of 1500–1700 MPa and Young’s modulus of about 40 GPa [108].

In contrast with other green composites, the reinforcing and matrix phases are both celluloses. Crystalline cellulose fibrils (filler) are reinforced in soft regenerated cellulose phase (matrix), has good mechanical properties due to the ideal fibril/matrix interfacial compatibility. These find their applications such as food packaging films, optical devices. Few researchers have also focused on synthesizing green fire-retardant chemicals and production of fire-resistant green composites. The inherent chemical composition of the proteins that contains large amount of nitrogen does not burn easily and these kinds of proteins can be used as reins in green composites and can produce fire retardant composites. In the recent decades all-cellulose composites have attracted significant interest [107].

Materials that incorporate functions such as sensing, actuation and control can be termed as smart materials. A lot of attention has been gathered around the development of smart polymer nanocomposites because of the fantastic improvements achieved in these materials. They are most widely used in applications such as sensor and actuator, stretchable electronic, wearables electronics, smart textiles, drug delivery, aircraft and aerospace applications [109]. One of the recent advancements in the composites industry is self-healing green resins and composites. Developing of microcracks in the polymeric composites can be life threatening in some cases. One of the innovative ways to cope with those cracks is to develop autonomously self-healing resins and composites that can heal microcracks as they are formed. Microcapsules that contain the healing agent are dispersed in the resin or polymer, the rupture of the microcapsules will release the healing agent and it will fill the microcrack and a bridge will be formed between two microcrack surfaces [109]. Functionally graded materials are novel materials whose properties change gradually with respect to their dimensions. These materials have obtained a huge attention due to their graded properties at every single point in various dimensions [110]. They are used mainly in the sectors such as aerospace, defense, nuclear industry, biomedical and electronics [111]. Functionalization of fibers using nanoparticles have attracted tremendous attention. One of the authors of the present work has a wide experience on the functionalization of flax [112,113] and jute fibers [114,115] by utilizing CaO [112], ZnO [113,116], Ag nanoparticles [114,115]. These kinds of novel materials have found their application in composites for military applications.

As a result of cost-performance benefits, composites become the choice of materials for several industries. Very significant innovations are expected in the years to come to meet the large-scale production and large penetration into the various advanced industries. The key trends in the field of composite materials are reducing the weight of parts in the areas of automotive and aerospace. In addition, advances improved the performance of the reinforcements and matrices for achieving higher performances of green composites, reduction in the cost of various composite parts, predictable technologies for the manufacturing of the composites in large scale. The composites are used in the various sectors that range from simple household to light and heavy industries. These are widely used due to desirable characteristics such as lightweight, corrosion resistance, good specific properties and flexibility in engineering the composites. It is anticipated that these composites will penetrate various industries shortly [117]. The factors such as durability compatibility, affordability and sustainability, including resource availability, land use, biodiversity, environmental impact are the challenges for converting renewable resources into industrial materials. The industry sectors and users of natural fibers are now focused in an eco-design approach. The main intention is to scale up the knowledge of fibers and preforms (materials for matrix and for reinforcements—long or short fibers, woven and non-woven fabrics, manufacturing technologies, prepreg composites, hybrid association of materials etc.) and attract the new industries [118].

Environmental concerns have resulted in many researchers looking for bio-based materials such as green composites. The natural fibers have emerged as new alternatives in place of manmade fibers such as glass fibers, aramid fibers and so forth. In this scenario, especially bast fibers are gaining interest in the field of high-performance natural fibers. These bast fibers include flax and jute and are immediately attractive as they possess comparable tensile strength and stiffness as that of glass fiber and contain favorable vibration damping and non-abrasive properties [119]. The intention of making the natural fibers to compete with synthetic fibers in terms of performance has resulted in a huge number of modifications to the natural fibers. These changes have improved the quality of natural fibers that resulted in emerging as a substitute for the traditional materials [117]. The emerging innovations with the use of natural fibers are improvement in strength and stiffness to compete with other fibers, enhancement of impact resistance, aesthetic properties and sustainability [120].

Despite being lighter than the synthetic fibers, natural fibers such as hemp, flax, jute and kenaf have desirable characteristics such as good strength and stiffness, low cost and biodegradable. Hence considering the advantages, natural fiber reinforced composites replacing the synthetic fiber reinforced composites is an emerging trend. Depletion of petroleum resources and the instability of crude oil prices as well as the desire to avoid the landfill disposal, the development of polymers from natural sources has received a considerable attention in the recent years. Polymers and natural fibers together can create a lightweight component for interior/exterior structural components. Moreover, in the future, bioinspired materials may replace traditional composites and can dominate the industry [117].

## 10. Conclusions

Due to the non-biodegradable nature of petroleum derived polymeric products, alternatives for synthetic fiber and resins are attracting the attention of the research community. Concerning this purpose, green chemistry and environmental engineering are regarded as the main opportunity for the expansion of the next generation of materials and processes. Many techniques to improve the performance of natural composites are constantly evolving, although this type of composites still exhibits lower structural capacity compared to synthetic fiber composites. The feasible use of biopolymers in composite products and their reinforcement with natural fibers requires a great effort from a scientific point of view.

This paper reported the recent advances that have been made in the field of biocomposites made of PLA matrices reinforced with both jute and flax fibers. The first topic addressed was the information regarding the processing of this type of composites. The most relevant production techniques has been described, namely, Compression Molding and Injection Molding and the different optimized parameters were detailed for the mixing of the formulations via extrusion (rotation speed and temperature) and for the molding of final composites (temperature, pressure and time). In addition, the factors that influence film-stacking production have been contemplated, as well as the fiber orientation and distribution.

The properties of flax reinforced PLA composites have been displayed and different solutions to improve the drawback of this type of composites were described. The conditions used to produce the composites and the different treatments applied reveal that PLA-Flax multilayered composites may achieve tensile strength and Young’s modulus values over 100 MPa and 10 GPa respectively. Also, the modification of the initial properties of both PLA and flax fibers may also lead to increased values of impact strength, energy absorption and thermal performance of the composites.

Related to PLA composites reinforced with jute, lower mechanical performance was found compared to PLA-Flax materials. However, the functionalization of jute fibers creates a possibility for the functionalization of the overall composites. Thus, apart from the improvement of durability and matrix-fiber interface, other properties may be incorporated into the composites, such as fire retardancy barrier.

## Figures and Tables

**Figure 1 polymers-12-02373-f001:**
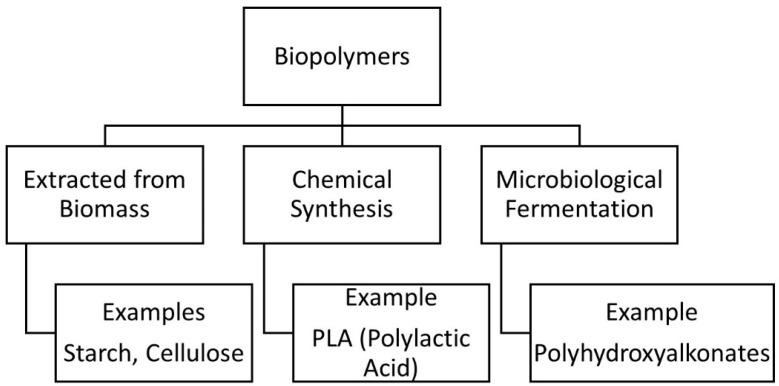
Classification of Biopolymers [4].

**Figure 2 polymers-12-02373-f002:**
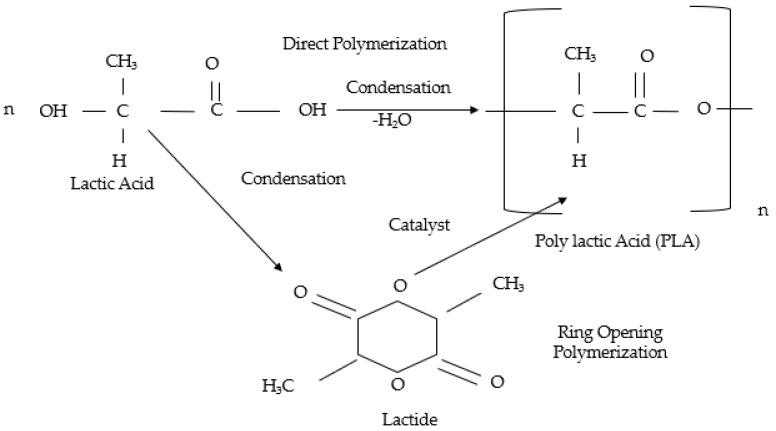
Polylactic Acid (PLA) Polymerization [14].

**Figure 3 polymers-12-02373-f003:**
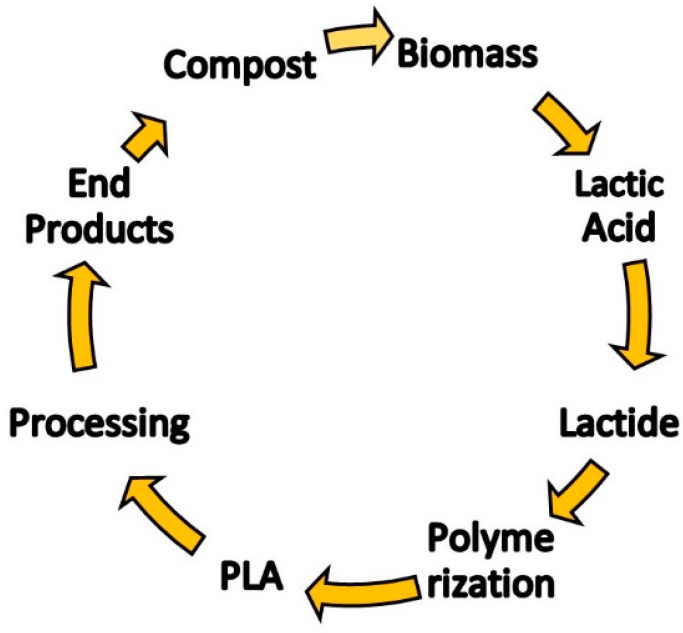
Life Cycle of PLA Biopolymer [19].

**Figure 4 polymers-12-02373-f004:**
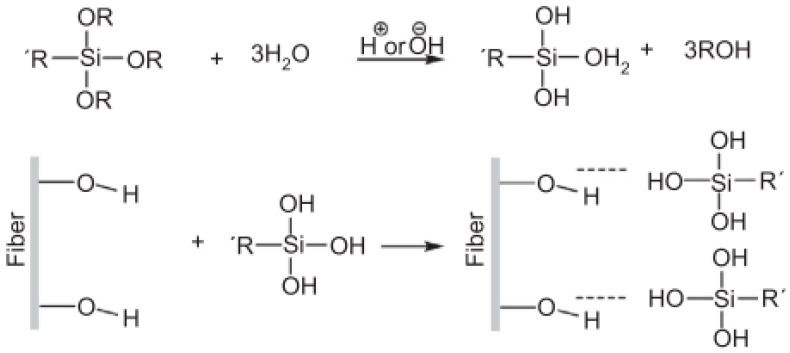
Reaction between the silane groups and the functional groups of natural fibers [48]. This article was published in Surface Modification of Natural Fibers in Polymer Composites, Ferreira, D.P.; Cruz, J.; Fangueiro, R., Page No. 17, Copyright Elsevier (2019). Elsevier Ltd: Amsterdam, The Netherlands, 2019.

**Figure 5 polymers-12-02373-f005:**
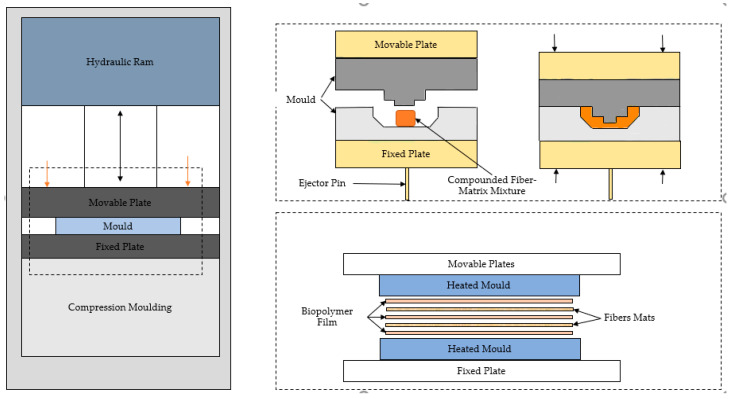
Schematic diagram of compression molding Adapted From Reference [81].

**Table 1 polymers-12-02373-t001:** Physical properties of PLA and PLLA [24].

Biopolymers	Density (g/cm^3^)	Tensile Strength (MPa)	Youngs Modulus (GPa)	Glass Transition Temperature (°C)	Melting Temperature (°C)
PLA	1.21–1.25	21–60	0.35–3.5	45–60	150–162
PLLA	1.24–1.30	15.5–150	2.7–4.14	55–65	170–200

**Table 2 polymers-12-02373-t002:** Comparison of mechanical properties of PLA and synthetic polymers.

Biopolymers	Tensile Strength (MPa)	Youngs Modulus (GPa)	Elongation at Break (%)	Reference
PLA	53	2.4	5	[15]
Polypropylene	31	1.5–2	80-350	[15]
HDPE	14.5–38	0.4–1.5	2.0–130	[25]
Polystyrene	25–69	4–5	1–2.5	[25]
Polyamide	56–90	2	70	[15]

**Table 3 polymers-12-02373-t003:** Classification of Natural Fibers Adopted From [2].

Bio Fibers	Sources	Examples
Animal Fiber		Wool/hair—sheep, camel, rabbit hair goat hair, yak, horsehair, Silk Asbestos
Mineral		Asbestos, wollastonite
Plant fiber	Stalk fiber	Bamboo, wheat, rice, grass, barley, corn, maize, oat
	Fruit fiber	Coconut, betelnut
	Seed fiber	Cotton, oil palm, kapok, alfalfa
	Leaf fiber	Sisal, banana, palm, date palm, pineapple, henequen, agave
	Bast fiber	Hemp, jute, banana, flax, kenaf sugarcane, ramie, roselle

**Table 4 polymers-12-02373-t004:** Chemical composition of natural fibers.

Fibers	Cellulose (wt %)	HC (wt %)	Lignin (wt %)	Pectin (wt %)	Ash (wt %)	Waxes (wt %)	MC (%)	MFA (Deg)	Ref.
Abaca	56–63	20–25	7–9	-	3	3	5–10	-	[31]
Bagasse	55.2	16.8	25.3	-	1.5–5	-	8.8	-	[33]
Bamboo	73.83	12.49	10.15	0.37	9.6	-	3.16	-	[31]
Banana	60–65	6–8	5–10	-	2.7–10.2	-	10–15	11	[34]
Coir	32–43	0.15–0.25	40–45	-	-	-	10–12	30–39	[33]
Cotton	82.7	5.7	-	-	-	-	1	20–30	[31]
Curauá	73.6	9.9	7.5	-	3.9–9.6	-	-	-	[33]
Date-palm	30.3–33.5	59.5	27–31.2	-	5	-	-	-	[35]
Elephant Grass Grass	45.6	-	17.7	-	-	-	-	-	[36]
Flax	71	18.6–20.6	2.2	2.3	-	1.7	8–12	5–10	[33]
Hemp	68	15	10	1	0.8	0.8	6.2–12	2–6.2	[33]
Henequen	77.6	4–8	13.1	-	-	-	-	-	[29]
Jute	61–71	14–20	12–13	-	0.8	0.5	12.5–13.7	8	[33]
Kenaf	45–57	21.5	8–13	3–5	2–5	-	-	-	[29]
Oil-palm	65	-	29	-	2.4	-	-	46	[33]
Pineapple	70–80	18.8	12.7	1.1–1.2	0.9–1.2	3.2–4.2	11.8	8–15	[29]
Ramie	68.6–76.2	13–16	0.6–0.7	1.9	-	0.3	7.5–1.7	7.5	[29]
Rice Husk	35–45	19–25	20	-	-	14–17	-	-	[33]
Rice Straw	41–57	33	8–19	-	14–20	8–35	6.5	-	[33]
Sisal	65	12	9.9	10	0.6–1	2	10–12	10–22	[33]
Sugar Palm	53.41	7.45	24.92	-	4.27	-	8.7	-	[37]
Wheat Straw	38–45	15–31	12–20	-	6.8	-	10	-	[33]

HC—Hemicellulose; MFA-Micro-fibrillar Angle; MC—Moisture Content.

**Table 5 polymers-12-02373-t005:** Physical and mechanical properties of natural and synthetic fibers.

Fibers	Density (g/cm3)	Tensile Strength (MPa)	Youngs Modulus (GPa)	Elongation (%)	Moisture Absorption	Ref.
Abaca	1.5	410–810	41	1.6	-	[13]
Banana	1.35	721.5–910	29	2	-	[33]
Chicken Feathers	0.89	100–200	3–10	-	-	[13]
Coconut	1.15	131–175	4–6	15	-	[13]
Coir	1.2	175–220	4–6	15–30	10	[13]
Cotton	1.5–1.6	287–597	5.5–12.6	3–10	8–25	[13]
Curaua Leaf	-	1250–3000	30–80	-	-	[40]
Flax	1.4–1.5	345–1500	27.6–80	1.2–3.2	7	[13]
Harakeke	-	440–990	14–33	-	-	[41]
Hemp	1.48	550–900	70	1.6	8	[13]
Henequen	1.4	500	13.2	4.8	-	[33]
Jute	1.3–1.46	393–800	10–30	1.5–1.8	12	[13]
Nettle	1.51	650	38	1.7	-	[33]
Pineapple Leaf	1.07–1.50	413–1627	34.5–82.5		11.8	[13]
Ramie	1.5	220–938	44–128	2.0–3.8	12–17	[13]
Sisal	1.33–1.5	400–700	9–38	2–14	11	[13]
Spartium Juncem L	1.55	986.46	17.86	-	-	[41]
Softwood	1.5	1000	40.0	-	-	[13]
E-Glass	2.5	2000–3500	70.0	2.5–3	-	[13]
S-Glass	2.5	4570	86.0	2.8	-	[13]
Aramide (normal)	1.4	3000–3150	63–67	3.3–3.7	-	[13]
Carbon (standard)	1.4	4000	230–400	1.4–1.8	-	[13]

**Table 6 polymers-12-02373-t006:** Advantages and disadvantages of natural fibers [38,39].

Advantages	Disadvantages
Lower specific weight results in a higher specific and stiffness than glass	Low mechanical properties especially impact resistance
Renewable resource	Moisture sensitivity
Production with low investment	Low thermal stability
Low abrasion and hence less tool wear	Low durability
Abundantly available	Poor fire resistance
Biodegradable	Poor fiber-matrix adhesion

**Table 7 polymers-12-02373-t007:** Processing parameters for compression molding (PLA/Flax) composites.

Method	Rotation Speed (RPM)	Temperature (°C)	Process	Mold Temp (°C)	Pressure (MPa)	Time (Min)	wt %	Ref.
TSE	150	154–171	CM	185	-	-	30	[56]
FS	-	-	CM	185	4	-	22	[57]
MC	100	180	CM	180	30	10	30	[58]
SC	-	-	CM	100	5	30	-	[59]
CS	-	-	CM	180	2	5	49.5	[60]
AL	-	-	CM	180	4	5	48	[60]
FS	-	-	HP	180	0T, 2T, 1T, 0.5T, 6T	5, 1.5, 1, 0.5,	30	[61]
FS	-	-	HP	124	5	6 h	22	[62]
TSE	150	170–175	-	-	-	-	-	[63]
SC	-	-	HP	180	-	5	49.3	[64]
SC	-	-	HP	180	-	5	60.7	[64]
SC	-	-	HP	180	34	5	34	[65]

AL—Air Laying; CM—Compression Molding; HP—Hot-Pressing; CS—Conventional Spinning; FS—Film Stacking; MC-Micro Compounder, SC—Stacking; TSE-Twin Screw Co-Rotating Extruder; Wt—Weight.

**Table 8 polymers-12-02373-t008:** Processing parameters for hot press and injection molding (PLA/Flax).

Process	Temperature (°C)	Pressure (MPa)	Process	Temperature (°C)	Pressure (MPa)	wt %	Ref.
HP	170	18	IM	170, 175, 180	10	10, 20, 30	[66]
HP	170	18	IM	170, 175, 180	12	40	[66]

HP—Hot Pressing; IM—Injection Molding.

**Table 9 polymers-12-02373-t009:** Processing parameters for extrusion and injection molding (PLA/Flax).

Equipment	Rotation Speed (RPM)	Temperature (°C)	Process	Melting Temperature (°C)	Mold Temperature (°C)	wt %	Ref.
TSC	100	180	IM	180	25	25	[67]
TSE	-	180	IM	200	25	10, 30	[68]
TSE	150	172–177	IM	177–182	55	2.5–12.5	[69]

IM—Injection Molding; TSC—Twin Screw Compounder; TSE—Twin Screw Extruder.

**Table 10 polymers-12-02373-t010:** Processing parameters for compression molding (PLA/Jute) composites.

Method	Rotation Speed (RPM)	Temperature (°C)	Process	Mold Temperature (°C)	Pressure (MPa)	Time (Min)	wt %	Ref.
WM	-	-	CM	185	1.33	8	37.3	[70]
-	-	-	CM	180	6.89	40	50	[71]
FS	-	-	CM	180	13.79	40	50	[71]
-	-	-	CM	80–130	350 KN	180	-	[72]
			CM	150	0.6	10	40	[73]
Extrusion	25	75 and 80	HP	165	-	2–3	3–15	[74]
-	-	-	HP	110	50	180		[75]

CM—Compression Molding; FS—Film Stacking; HP—Hot Press; KN—Kilo Newtons; WM—Winding Machine.

**Table 11 polymers-12-02373-t011:** Processing parameters for extrusion and injection molding (PLA/Jute).

Method	Rotation Speed (RPM)	Temperature (°C)	Process	Temperature (°C)	Mold Temperature (°C)	wt %	Ref.
-	-	-	IM	160	25	50	[76]
TSE	150	180	IM	180	-	-	[77]
TSCE	80	155–170	IM	170–190	40–50	15	[78]
TSCE	80	155–170	IM	170	30	15	[79]
TSE	100	160–170	IM	175–180	-	10	[80]

IM—Injection Molding; TSCE—Twin Screw Co-Rotating Extruder; TSE—Twin Screw Extruder.

**Table 12 polymers-12-02373-t012:** Mechanical properties of PLA composites reinforced with flax fibers.

Fiber Content wt (%)	Process	Tensile Strength (MPa)	Flexural Strength (MPa)	Impact Strength (kJ/m^2^)	Young’s Modulus (GPa)	Additional Information	Ref.
	IM	74.3	-	12.7	-	ST and PC	[67]
22	FS	99.0	140	-	16.0	UT	[62]
22	FS	102.5	117	-	12.5	ST	[62]
22	FS	63.0	83	-	12.7	TBCT	[62]
22	FS	35.0	63	-	25.0	MAHT	[62]
30	HP	177	-	-	10.8	Before conditioning	[61]
30	HP	187		--	12.2	Nano silica coated and before conditioning	[61]
30	HP	89	-	-	4.68	After conditioning	[61]
30	HP	114	-	-	6.2	Nano Silica coated and after conditioning	[61]
30	CM	55.4		-	2.90	Coated with PDA film	[58]
30	CM	60.1	-	-	3.40	-	[58]
48	CM	83.0	130.0		9.3	Random	[60]
48	CM	151.0	215.0		18.5	Aligned	[60]

CM—Compression Molding; FS—Film Stacking; HP—Hot Pressing; IM—Injection Molding; MAHT—Maleic Anhydride Treated; PC—Polymer Coated; ST—Silane Treated; TBCT—Tributyl Citrate Treated; UT—Untreated.

**Table 13 polymers-12-02373-t013:** Mechanical properties of PLA composites reinforced with jute fibers.

Fiber Content wt (%)	Process	Tensile Strength (MPa)	Flexural Strength (MPa)	Impact Strength (kJ/m^2^)	Young’s Modulus (GPa)	Additional Information	Ref.
50	IM	62.2	98.8	2.21	-	-	[76]
-	IM	90.7	-	4.22 (kJ/m)	12.3	Short Fiber Pellet mix	[77]
-	CM	32.3	41.8	3.5 J	2.11	Woven Jute Fabric	[72]
30	IM	71.7	-	-	9.15	Lignin 4%	[95]
-	RFM	53.16	128.3	-	8.11	-	[75]
20	IM	55	110	1.6	1.7	10% NaOH + H_2_O_2_	[99,100]
25	IM	54	80	2.5	2.7	05% NaOH + H_2_O_2_	[99,100]
30	IM	70	-	26 (J/m)	-	Alkali Treated	[101]
30	IM	80	-	28(J/m)	-	Alkali + Silane Treated	[101]
15	IM	50	78	4.8	4.8	Treated with DOPO-ICN	[78]
10	-	61.71	93.68	19.26	4.2	Graphene Modified with DOPO	[102]
5	SC	70.30	-	-	3.30	PLA Composite Films with Jute Nano Fibers	[103]
3	SC	69.80	-	-	3.20	NaOH Treated Jute Nano Fibers	[104]
40	HP	77.5	115.3		3.67	Core Shell Nanoparticles (SiO_2_–PBA–NH_2_)	[105]

CM—Compression Molding; DOPO—9,10-dihydro-9-oxa-phosphaphenanthrene-10-Oxide; DOPO—ICN-Phosphorus Based Compound; HP—Hot Pressing; IM—Injection Molding; RFM—Resign Film Method; SC—Solvent Casting.

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
