# Peer review of "PLA Composites Reinforced with Flax and Jute Fibers—A Review of Recent Trends, Processing Parameters and Mechanical Properties"

_polymers, 2020, doi:10.3390/polym12102373_

Round 1

Reviewer 1 Report

Nice review and timely topic.

Avoid conjunction words at the start of sentences, e.g. Besides (line 18), While  (line 23)

Mixture of numbered (as per reference list) and Harvard style referencing in the text!

Line 16: “environmental” or “sustainability” rather than “conservational”?

Line 36: “phases”, not “phased”.

Line 62: Not just in the past as straw-reinforced earth building is a traditional craft still practised in Iberia (adobe), Brittany (bauge) and England (cob).

Line 106: extend “industrial composting” to differentiate between aerobic vs anaerobic composting, and £35°C vs > 50°C?

Line 113; “not”, not “never”.

Line 120: (Error! Reference source not found.)

Line 125: State that PLA is the only (?) commercially available polymer with glass transition temperature above ambient and melting point below the degradation temperature of lignocellulosic fibres.

Lines 162 and 474: not normal to include cited author initial.

Line 179: “Error! Reference source not found” twice!

Lines 183, 427 and 442: capital “Y” for Young’s modulus as it is named after a person.

Line 283: Are coupling agents appropriate for natural fibres available which do not use silicon compounds that might produce harmful nanosilicate particles when incinerated at end-of-life?

Line 289: delete the “g” from “resign transfer method”

Line 290: the focus here is melt processing of the polymer.  There scope for in situ polymerisation during composite manufacture, e.g. https://doi.org/10.3390/polym11020339 and https://hal.univ-lille.fr/hal-02924962.

Line 297: delete “was” at “was as it”

Line 329: delete first “matrix” at “matrix fiber-matrix mixture”

Line 331: “blinders”?

Lines 335-336: How is the Bledzki et al PP-wood reference relevant to this paper?

Line 341 “heated in an oven” to what temperature?

Line 350: “40 wt.% jute fibre mats” presumably refers to the final proportion in the composite?

Line 366: delete “with” from “reinforced with PLA”

Line 367: “were” not “was”

Line 388: quote strengths and moduli at MPa/GPa (with standard deviation) respectively, then as %?  Similar considerations apply at other points in the manuscript where reference values are not stated (100% increase of very poor values is still poor!)!

Line 401/402/403: “saline” or “silane”?

Line 436: does “decrease in water uptake by more than 10 %” mean the water uptake was >>10% or water uptake is now ~90% of the untreated case?

Line 438: LOI is Limiting Oxygen Index?

Line 440: What is “Fep-Flax”?

Line 445: why is Random abbreviated to “NM”?

Line 568: IFSS – Inter-Facial Shear Strength?

Line 580: “with a[n] electron beam”

Lines 587-588: “was also improved was achieved”!

Lines 596-600: round the percentages to integer values!

Line 598: “while [composites with a xx wt. % of NF] showed”

Line 608: NFPC?

Line 626: “re[s]ins”

Line 667: “bast” not “blast”

Line 670: “dampening” is making wet. “damping” is vibration suppression”

Line 695: I think “all” is overstating the case?
Line 695: “recent” rather than “last” which implies no further advances will occur!

Reference 5: author is just initials!  No need for title in all capitals!

Author Response

Dear Sir/Madam,

I would like to thank you for your review and suggestions. Please see the attachment

Thanks and Regards,

Usha Kiran

Reviewer 2 Report

The review manuscript by Sanivada U.K. et al. “PLA Composites Reinforced with Flax and Jute Fibres: A Review of Recent Trends, Processing Parameters and Mechanical Properties” presents current state of the art of PLA based composites with flax and jute as natural reinforcement. The paper is well organized and gives systematic overview on presented subject. This manuscript can be accepted after some minor revisions.

Minor comments:

  1. In case of the Author cited papers [112-116] it should be more clearly pointed, that this are their works.
  2. Please check the space character in all paper; e.g. double space in lines: 55,62, 314.
  3. Lines 120 and 179 Reference error.
  4. Please correct the numbers of Table 8 and 9 (now is 3 and 4).

Author Response

Dear Sir/Madam,

I would like to thank you for your review and for your suggestions.

Minor comments:

1. In case of the Author cited papers [112-116] it should be more clearly pointed, that this are their works.

It was mentioned in the document

2. Please check the space character in all paper; e.g. double space in lines: 55,62, 314.

Removed double spaces

3. lines 120 and 179 Reference error.

Table with number is added

4. Please correct the numbers of Table 8 and 9 (now is 3 and 4).

The numbers are changed to 8 and 9

Thanks and Regards

Usha Kiran